# Ignition Delay Time Modeling in Wire-EDM

**Paulo Matheus Borges Esteves** *,† , **Micha Hensen** † , **Michal Kuffa and Konrad Wegener**

ETH Zürich, Leonhardstrasse 21, 8092 Zürich, Switzerland; hensenm@student.ethz.ch (M.H.);
kuffa@iwf.mavt.ethz.ch (M.K.); wegener@iwf.mavt.ethz.ch (K.W.)
* Correspondence: borges@iwf.mavt.ethz.ch
† These authors contributed equally to this work.

**Abstract:** This study presents a comprehensive investigation and modeling of the ignition delay time ($t_d$) in wire-EDM (WEDM). The research focuses on the influence of gap distance, discharge energy, and piece height on the stochastic distributions of $t_d$, providing important insights into the complex properties of these distributions. Observations indicate that these parameters exert significant yet intricate influences on $t_d$, with a particular emphasis on the gap distance. A critical value was identified, around 8 μm to 10 μm, that divides the stochastic behavior. To capture the binomial nature of $t_d$, a mixture probability model consisting of two Weibull distribution curves was developed and validated through extensive experimentation and a data analysis. The model demonstrated strong agreement with observed cumulative probability curves, indicating its accuracy and reliability in predicting $t_d$. Further, a sensitivity analysis revealed regions of fast change, emphasizing the challenges and importance of careful parameter selection in control of WEDM processes. The findings of this study contribute to a deeper understanding of WEDM processes and provide a modeling approach for predicting $t_d$. Future research directions include refining the model by incorporating additional input parameters, investigating the influence of other process variables on $t_d$.

**Keywords:** electrical discharge machining (EDM); ignition delay time; wire-EDM (WEDM); Weibull distribution

## 1. Introduction

Wire Electrical Discharge Machining (Wire-EDM) represents a non-traditional, sophisticated manufacturing technique characterized by the application of electrical discharges for machining conductive workpieces immersed in a dielectric medium, typically deionized water. In the process, electrical discharges are produced in the interface between the wire electrode and the workpiece, both separated by a gap of a few micrometers containing the dielectric medium. Depending on the characteristics of voltage over time in this interface, the discharges are generally categorized into normal discharges or short circuits. Normal discharges facilitate a stable cutting environment and minor roughness, whereas short circuits and arcing result in adverse effects. The classification of discharges is important for an optimal understanding and control of the Wire-EDM process.

This classification is done through the evaluation of the voltage behavior before, $t_d$, and during the discharge. Liao et al. [1], for example, have developed a pulse discrimination system for Wire-EDM. They have studied, among others, the distribution of $t_d$ of normal discharges during a specific sampling period. Moreover, Qin et al. [2] have developed a real-time gap state discrimination method to enable stable processing.

The precise mechanism that drives the dielectric fluid breakdown in EDM is still an area of ongoing scientific inquiry. Multiple theories have been proposed to shed light on the process leading to a discharge, incorporating ideas like the formation of gas bubbles via electrolysis and subsequent plasma generation within these bubbles, the inception of charge avalanches, and the establishment of particle chains [3–5]. However this phenomenon takes place, it has a strong connection to $t_d$, which is a critical parameter in managing the

Wire-EDM process. The $t_d$ is commonly used to estimate the conditions within the working gap, and the behavior of EDM systems [2,6]. Morimoto and Kunieda [7] have studied the effect of the gap width, concentration of debris particles, and the machining area on $t_d$ for die-sinking EDM. They have quantified the influences using Laue plots, which supposes the assumption of exponential behavior for $t_d$. They have also observed that the $t_d$ for single pulses is distinctly longer compared to actual machining. In a follow up work [6], they have simulated the die-sinking EDM process using an exponential distribution for the $t_d$, shown in Equation (1).

$$t_{d,ave} = 8.2 \cdot 10^{12} \left( \frac{g^{8.8} \cdot r^{2.9}}{a^{1.2} \cdot c^{1.6}} \right) \tag{1}$$

where $t_{d,ave}$ is the average ignition delay time and $c$, $g$, $r$ and $a$ are the concentration of debris particles, gap width, debris particles diameter, and machining area, respectively.

Other researchers have also investigated the influences on discharge location and $t_d$. This includes the debris concentration in the dielectric medium [3,8], the pause time between consecutive discharges, and the spatial distance between the tool and workpiece [4,9].

An abundant amount of research goes into the implications of discharge intensity on multiple aspects of Wire-EDM, like its impact on the surface texture of used wires [10], workpiece surfaces [11,12], surface roughness [13,14], and the size and volume of debris produced by discharge [15]. But the stochastic aspect of discharge generation has not been thoroughly studied.

In more recent work, such as that by Wang et al. [9], $t_d$ measurements in Wire-EDM have been conducted, investigating the effect of surface roughness, gap distance, and flushing pressure in continuous machining. However, a model that predicts the factors impacting the discharge type, mean $t_d$, and its distribution in Wire-EDM is notably missing. Developing a reliable theoretical framework requires the collection of a significant quantity of data and a thoughtful consideration of the interrelationships among the various factors.

The present study aims to fill this gap in the current literature by systematically exploring the factors affecting discharge type, mean $t_d$, and its distribution in Wire-EDM, all the while taking into account the complex interplay among the explaining factors. This study gathers statistically significant data and constructs a comprehensive model capable of predicting spark distribution across numerous discharges. Such a model could also prove to be useful in the future for optimizing the process and for technology choice.

## 2. Materials and Methods

### 2.1. Instrumentation

The experiments were conducted on an AgieCharmilles CUT E 350 Wire-EDM machine, which is a well-established and widely used electrical discharge machining platform. The purpose of these experiments was to investigate the discharge delay time, $t_d$, in the EDM process, which plays a crucial role in process control and optimization. To accurately record the voltage levels of the wire during the machining process, a LECROY WaveRunner 604Zi oscilloscope was employed. Two cables connected the wire contacts to the oscilloscope, while one additional wire was connected to the machine table, recording the ground potential. Figure 1 presents a scheme of the voltage curve acquisition setup.

### 2.2. Machine Setup and Materials

Three series of experiments were conducted using workpieces made from 1.2379/X155CrVMo12-1 steel blocks, which is a widely used material in the manufacturing industry. Blocks of different heights were prepared to assess the effect of workpiece height on the discharge delay time. Prior to experimentation, the EDM machine was meticulously prepared to ensure accurate measurements and reliable outcomes. The workpieces were carefully cut and trimmed using the EDM machine itself, achieving surfaces with minimal roughness, $R_a$ of 0.3 μm. This step was crucial to provide a consistent and well-defined experimental platform. The workpieces are left clamped throughout the entire experiment

sequence, reducing potential alignment issues and enabling the use of the internal machine scales for accurate measurement. Table 1 summarizes the expected maximum surface deviation, given by the machine manufacturer, and the surface roughness.

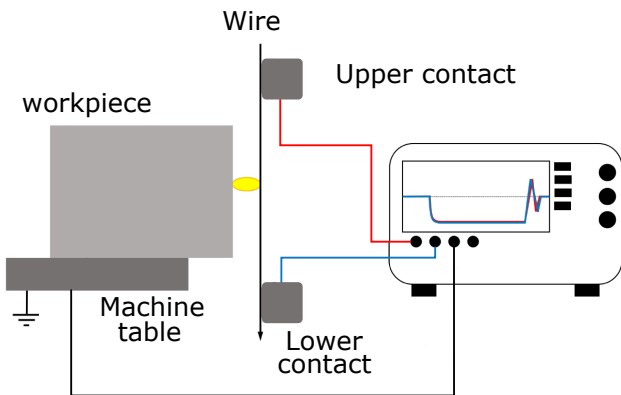

**Figure 1.** EDM experimental setup to record Voltage curves, illustrating the connections between the machine, workpiece, and oscilloscope.

**Table 1.** The maximum expected surface deviation and roughness for various workpiece heights.

| Piece Height | Flatness Tolerance | Surface Roughness ($R_a$) |
| :---: | :---: | :---: |
| 5 mm | 1.2 μm | 0.3 μm |
| 10 mm | 1.4 μm | 0.3 μm |
| 20 mm | 1.9 μm | 0.3 μm |
| 50 mm | 2.8 μm | 0.3 μm |
| 100 mm | 3.8 μm | 0.3 μm |

To ensure the reliability and reproducibility of the experiments, several measures were taken. The consumables, such as the ionic resin, the filters, and the electric contacts, were replaced before the start of the experiments. The conductivity of the water was closely monitored and maintained within the range of 9–11 μS/cm during the preparation of the specimens and during the experiments. Additionally, the same spool of 0.2 mm of brass wire, produced by Thermocompact, was used through all experiments to prevent unexpected influences. To minimize the wire vibrations, the wire tension is set to 20 N.

The oscilloscope's setup was configured to use a timescale of 20 ms/div and a sample rate of $10^6$ samples per second. This allowed for the measurement of $t_d$ values up to a maximum of 180 ms. Under these conditions, the acquisition has a inaccuracy of 1 μs in time and 1 V in voltage. For the measurements, the wire travels parallel to the workpiece surface at a defined distance, as in a typical trim cut.

For the data acquisition, the wire travels at least 5 mm; this distance is punctually adjusted to have at least 400 acquisitions per experiment, which would allow a high statistical robustness. The wire travels at the maximum speed allowed: 40 mm/min. At the end of the experiment, the wire is moved forward by 0.5 mm to avoid any potential influence from the previous discharges. Then, new parameters are loaded and a new section is acquired.

Finally the waiting time between subsequent pulses was set to 409 μs, the maximum allowed by the machine, which should be ample time to deionize any remaining free ions and give some time to flush away debris generated in the last discharge.

### 2.3. Data Classification and Discharge Delay Time Measurement

Recorded voltage curves are classified into short circuits and normal discharges, as the discharge delay time is exhibited only by normal discharges. Subsequently, the beginning and end of the open voltage are defined to determine the duration.

The classification of the curves is done with the help of a Principal Component Analysis (PCA), which is a dimension reduction technique. Figure 2 shows the data of one experiment reduced to three dimensions. It is rather easy to see two clusters, one containing short circuits and the other normal discharges. Through a simple K-means clusterization, the voltage curves are then classified.

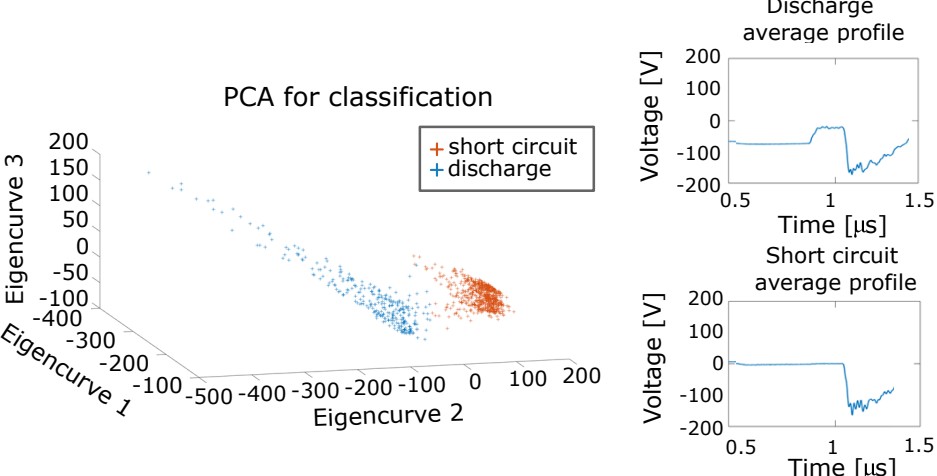

**Figure 2.** Principal Component Analysis (PCA)−driven dimensionality reduction and K-means clusterization of EDM voltage curves into short circuit and discharge.

The process of extracting $t_d$ from the curve involves identifying the beginning and the end of the open voltage. The open voltage is set at −85 V.

The start of the open voltage is determined by identifying the moment that the voltage goes below −35 V, while the end of the open voltage is defined as the first point where the curve crosses upwards above −70 V. This way, a $t_d$ can be determined for each voltage curve acquired.

## 3. Gap Variation

The gap variation experiment is done because the gap distance is believed to be the most important parameter for $t_d$. In the literature, the strong influence of the distance between the two electrodes on the $t_d$ distribution is noted [3,4,6,7,9]. With this experiment, some basic concepts for the modeling can also be established.

The experimental setup is done with a $h = 10$ mm high piece, along with $i = 17.9$ mJ of spark energy. The gap distance was varied from $d = 2$ μm to $d = 16$ μm, and the standard data cleansing steps were applied to classify the voltage curves and extract the $t_d$ for the normal discharges.

Firstly, the discharge types in relation to the gap distance: the results show a clear trend in the distribution of discharge types with respect to the gap distance. For gap distances of 2–4 μm, the percentage of normal discharges is below 2.5%, which could imply some difficulties in modeling $t_d$ at this range. However, as the gap distance increased, the percentage of normal discharges rapidly increased, reaching over 97.5% for a gap distance of 10 μm. After a gap distance of 15 μm, all discharges were normal discharges. Figure 3 illustrates this relationship.

After 20 μm, there are few sparks, i.e., the open voltage extends through the whole 900 μs window and no spark takes place.

Secondly, at around a gap distance of 8 μm to 10 μm, a change in behavior occurs. When plotting the $t_d$ distributions in histograms for gap distances of 3–8 μm, the probability distribution tends to have a decreasing function with a peak in the lower time interval range, as shown in Figure 4 on the left.

With a gap distance of 10 μm or more, the behaviour of the system suddenly changes to a clear bimodal distribution, which means a probability curve that has two distinct peaks.

Figure 4 on the right illustrates this. The first probability peak is similar to the lower gap width range, and a more prominent probability peak surges at 100–175 μs. With a further increasing gap distance, the first peak becomes less prominent, but it never fully disappears, even at a gap distance of 16 μm.

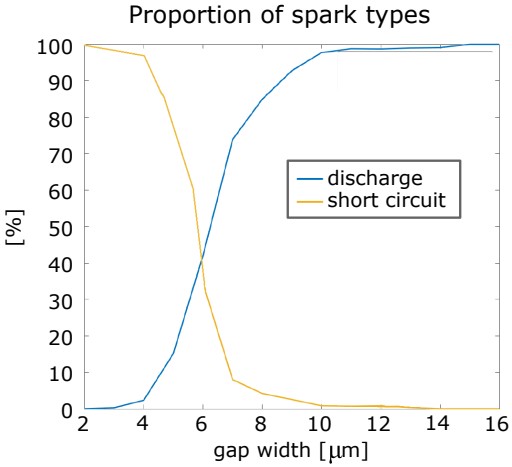

**Figure 3.** Distribution of discharges and short circuits as a function of gap distance.

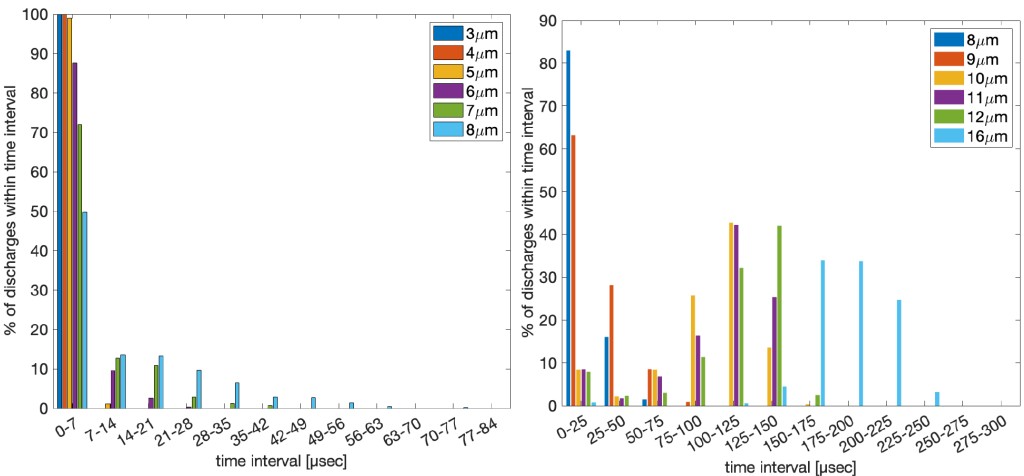

**Figure 4.** Histogram of the recorded ignition delay time over gap distances from 3 μm to 8 μm on the **left**, and from 8 μm to 16 μm on the **right**.

## 4. Factorial Design

A full factorial analysis was conducted to investigate the combined influences of the spark energy, gap distance, and piece height. Such a design allows for a thorough exploration of the entire experimental space.

The objective is to develop a numerical model to predict the behavior of the ignition delay time across the full range of machine parameters.

The experimental parameters and their respective levels are listed in Table 2.

**Table 2.** Parameters and level for the factorial design.

| Parameter | Levels |
|---|---|
| Gap distance | 4 μm, 8 μm, 10 μm, 12 μm, 14 μm, 16 μm, 20 μm |
| Spark energy | 0.32 mJ, 1.14 mJ, 17.9 mJ |
| Piece height | 5 mm, 10 mm, 20 mm, 50 mm, 100 mm |

The gap distance is selected based on the knowledge that beyond 20 μm, nearly no sparks occur, and below 4 μm, predominantly short circuits are observed. The energy levels were chosen based on common cutting strategies related to a first, second, and third cut. The energy value, quantified in mJ, for each level is calculated via the numerical integration of the electrical current multiplied by the burning voltage. This calculation is performed across several hundred observations. The median of these integrative measurements is then utilized as the representative energy value for the respective level and an associated uncertainty can also be calculated. The uncertainty in the energy will be used in the validation section.

The experimental design is comprised of 105 experiments. Additionally, six experiments with random levels inside the search space were also conducted to validate the model and will be addressed in the next section.

In the process of collecting $t_d$ values across various parameter combinations, it became apparent that the underlying structure of the data exhibited a pronounced bimodal character. This observation underscores the necessity for a model of greater complexity to accurately capture these nuanced dependencies. Furthermore, it was noted in several instances that the voltage remained open for the entire observation window of 900 μs, without any spark being observed. This distinct pattern of behavior is illustrated in Figure 5.

A clear influence of gap distance on the shape of the captured distributions is observed. In general, the average $t_d$ increases with the gap increase. A critical gap value can be observed again, after which the behavior changes drastically from monotonic to bimodal. Figure 5 also presents this change between 8 μm and 10 μm.

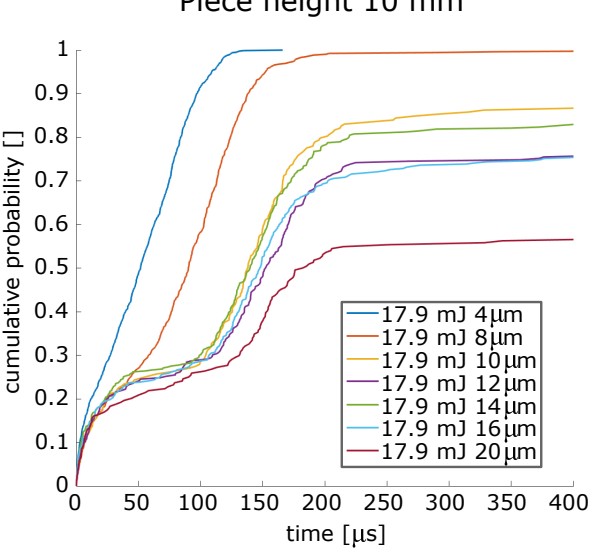

**Figure 5.** Experimental cumulative distribution function showing the influence of the gap on $t_d$.

The influence of the spark energy on the $t_d$ distributions is not clear. Figure 6 shows this difficulty, having the 1.14 mJ energy curve similar to the lower energy on the left and to the higher energy on the right. In general, it can be stated that the mean $t_d$ decreases with increasing intensity.

An influence of the piece height on $t_d$ is undeniable. On a general level, the smaller mean $t_d$ is found in the range between 10 mm and 50 mm. But it does not follow a clear general trend, as shown in Figure 7.

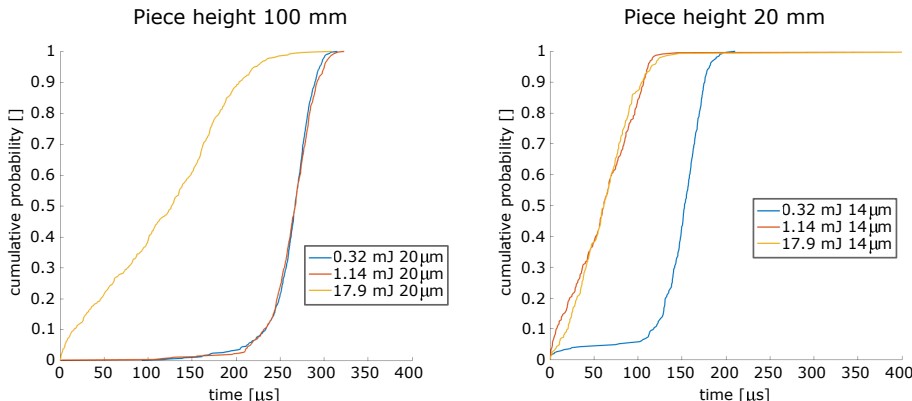

**Figure 6.** Experimental cumulative distribution function through different energies for piece height 100 mm and gap distance of 20 μm, on the **left**, and for piece height 20 mm and gap distance of 14 μm, on the **right**.

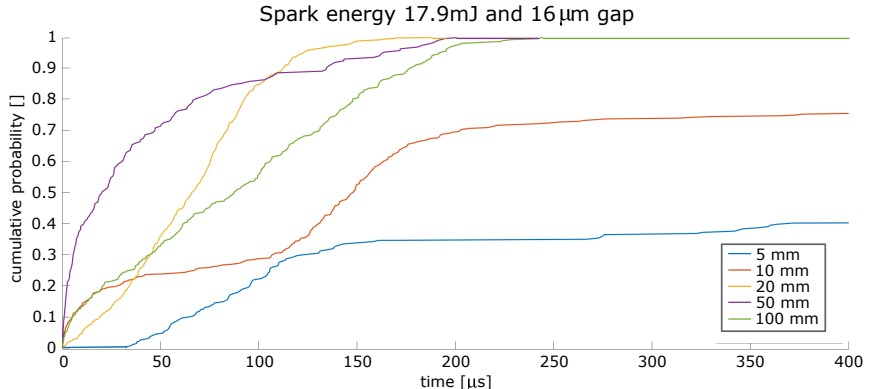

**Figure 7.** Experimental cumulative distribution function showing the influence of the piece height on the $t_d$-distribution.

## 5. Ignition Delay Time Modeling

To accommodate the bimodal distributions revealed by the previous observations, it is necessary to employ a mixture probability model. This study employs a superposition of two probability distributions to capture this behavior.

Weibull and exponential distributions are commonly used to describe $t_d$ in the literature [4–8,16]. According to the acquired data, an exponential distribution is inadequate due to its highest probability strictly at zero. Hence, modeling is continued using Weibull distributions.

The conceptualization of the superposition of two probability distributions are found in Equation (2):

$$t_d(t) = \alpha W_{b1}(\lambda_1, k_1, t) + (1 - \alpha)W_{b2}(\lambda_2, k_2, t). \tag{2}$$

In the above, $W_b$ represents a Weibull distribution with parameters $k$ and $\lambda$, both spanning $(0, \inf)$; $\alpha$ is the mixing parameter, constrained within $(0, 1)$.

An interpretation for the mixture distribution is that it characterizes two parallel failure modes of the dielectric.

Each of the describing parameters $(\alpha, \lambda_1, k_1, \lambda_2, k_2)$ is influenced by the gap distance, spark energy, and piece height.

Developing and fitting a model for each of the five parameters pose challenges due to the presence of numerous uncertain influences.

Numerous attempts were made to establish models using distinct functions to predict the parameters directly by studying the behavior of the recorded data relative to the input. Unfortunately, these were not successful. A different approach is necessary to tackle this problem.

Inspired by the modeling done by Morimoto and Kunieda [6], in which $t_d$ is modeled by the multiplication of the independent variables, the model presented here also aims to multiply the independent variables. To capture the complex behavior presented, the variables are transformed into functions.

This way, each of the describing parameters are written as a composition of separated functions, being each dependent on one single input variable. Considering $\lambda_1$ as an example, these describing parameters are rewritten as Equation (3):

$$\lambda_1(d, e, h) = g_d(d) \cdot g_i(i) \cdot g_h(h). \tag{3}$$

Here, $d$ is the gap distance in [µm], $i$ is the spark energy in [mJ], and $h$ is the piece height in [mm].

Even though their shape is unknown, they must assume a value at each observation point. Thus, $g_d$ must return a value at each of the seven gap distances. The same applies to the other two functions $g_i$ and $g_h$.

To find the values that those three functions should assume that best match each of the observations points, these values will support the choice for the shape for each function.

Figure 8 shows the found values for $\lambda_1$, for the three functions. In this case, an exponential function was chosen for $g_d$ and a second degree function was chosen for $g_i$. $g_h$ has shown an irregular pattern. Since no adequate function was found to model $g_h$, a linear interpolation with five supporting points is used. The choice was made keeping in mind that these values cannot be less than zero, else the Weibull becomes undetermined.

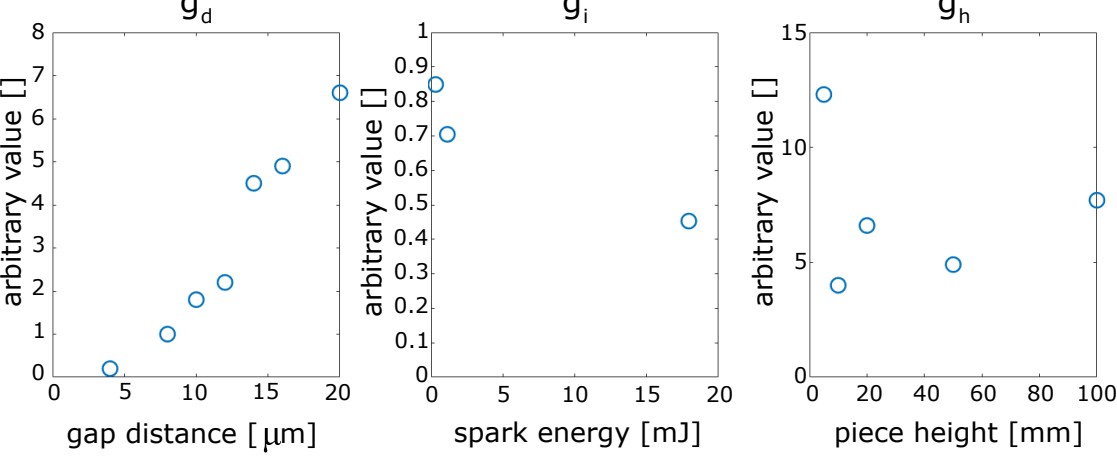

**Figure 8.** Values for the functions $g_d$, $g_i$, and $g_h$ composing $\lambda_1$ at the design levels.

The same procedure is done for the other functions $\alpha$, $b_1$, $a_2$, and $b_2$. The functions for the gap distance assume different shapes, while for the spark energy, a polynomial shape is used and for the piece height, a linear interpolation with five supporting points is used.

A genetic algorithm for the initial fitting was used, followed by a trust-region-reflective algorithm for the final solution, which is shown in Table 3:

**Table 3.** Functions that compose the $t_d$ model.

| Function | Gap Distance (d) | Spark Energy (i) | Piece Height (h) |
|---|---|---|---|
| $\alpha$ | $0.0122\,d^{2.21}$ | $0.0267\,i^2 + 0.657\,i + 2.19$ | 5 mm $\rightarrow 1.88 \cdot 10^3$ <br> 10 mm $\rightarrow 6.387 \cdot 10^{-2}$ <br> 20 mm $\rightarrow 2.237 \cdot 10^2$ <br> 50 mm $\rightarrow 2.929 \cdot 10^3$ <br> 100 mm $\rightarrow 2.248 \cdot 10^3$ |

**Table 3.** *Cont.*

| Function | Gap Distance (d) | Spark Energy (i) | Piece Height (h) |
|---|---|---|---|
| $a_1$ | $1.56\,e^{0.144\,d}$ | $0.445\,i^2 - 9.47\,i + 54.6$ | 5 mm $\rightarrow$ 1.856<br>10 mm $\rightarrow 9.426 \cdot 10^{-1}$<br>20 mm $\rightarrow 1.605 \cdot 10^{-1}$<br>50 mm $\rightarrow 1.026 \cdot 10^{-1}$<br>100 mm $\rightarrow 2.242 \cdot 10^{-1}$ |
| $b_1$ | $0.157\,e^{0.180\,d}$ | $-3.16\,i^2 + 55.8\,i + 58.7$ | 5 mm $\rightarrow 9.679 \cdot 10^{-3}$<br>10 mm $\rightarrow 7.164 \cdot 10^{-3}$<br>20 mm $\rightarrow 2.146 \cdot 10^{-2}$<br>50 mm $\rightarrow 1.970 \cdot 10^{-2}$<br>100 mm $\rightarrow 2.198 \cdot 10^{-2}$ |
| $a_2$ | $1640\,d^2 + 642\,d + 268$ | $-704\,i^2 + 34000\,i + 74.96$ | 5 mm $\rightarrow 5.001 \cdot 10^{2}$<br>10 mm $\rightarrow 3.826 \cdot 10^{-16}$<br>20 mm $\rightarrow 3.333 \cdot 10^{-1}$<br>50 mm $\rightarrow 2.208 \cdot 10^{2}$<br>100 mm $\rightarrow 2.559 \cdot 10^{2}$ |
| $b_2$ | $3.266\,10^5\,e^{-1.57\,d}$ | $88.03\,i^2 + 271\,i - 42.15$ | 5 mm $\rightarrow 9.799 \cdot 10^{2}$<br>10 mm $\rightarrow 4.985 \cdot 10^{2}$<br>20 mm $\rightarrow 4.367 \cdot 10^{-14}$<br>50 mm $\rightarrow 4.867 \cdot 10^{2}$<br>100 mm $\rightarrow 5.153 \cdot 10^{2}$ |

The gap distance is given in μm, spark energy is given in mJ, and piece height is given in mm.

Figure 9 shows the observed cumulative probability curves and the model prediction for a 100 mm high piece and 1.14 mJ spark energy for different gap distances.

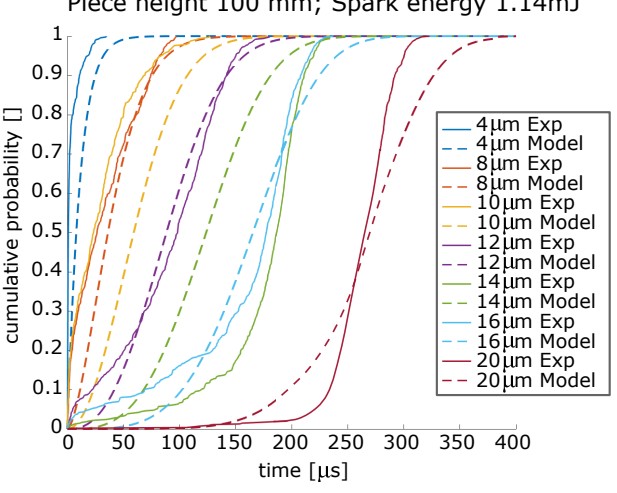

**Figure 9.** Comparison of modeled and observed distributions for a 100 mm high workpiece with a spark energy of 1.14 mJ.

Due to an uneven number of observations at each level, the error function was modified. It is not evaluated at every observation, but rather at every microsecond. This way, a normalization step is circumvented. The following Equation (4) shows the used error function:

$$f_{error} = \sum_{t=0\,\text{μs}}^{899\,\text{μs}} (e_{cdf}(t) - t_d(t))^2. \tag{4}$$

$e_{cdf}$ represents the experimental cumulative distribution function.

## 6. Sensitivity Analysis

Exploring the sensitivity of the model through a comprehensive analysis allows for the investigation of regions where changes in input parameters significantly impact the probability curves. By systematically evaluating parameter sensitivity, researchers can identify the most influential factors driving the model's response and gain critical insights into its behavior.

The sensitivity analysis involves varying one parameter at a time while keeping others constant, allowing for the isolation of individual parameter effects on the model's response. In this case, the gap distance and spark energy vary while keeping the piece height constant.

To do this analysis, the curves at a range of different energies and gap distances are determined. It is also necessary to define a scalar distance between the curves. It will be defined similar to the error function. The error is calculated as follows:

$$V = \sum_{t=0 \ \mu s}^{899 \ \mu s} (t_{d|i,d}(t) - t_{d|i+1,d+1}(t))^2 \tag{5}$$

where $t_{d|i,d}(t)$ is the ignition delay time probability for a given energy and gap, while $t_{d|i+1,d+1}(t)$ is the ignition delay time probability at the next energy and gap step. Figure 10 shows the sensitivity maps for changes of discharge energy and gap distance for each piece height.

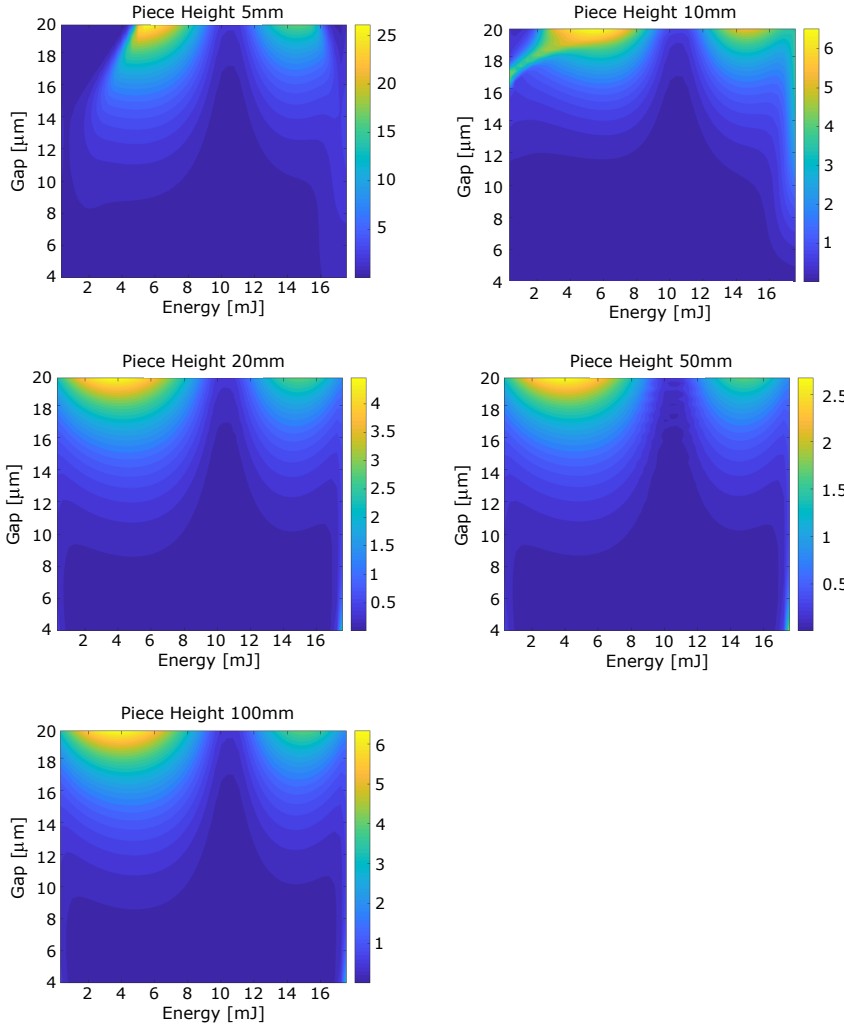

**Figure 10.** Sensitivity analysis showing regions of significant changes in $t_d$ in response to variations in gap distance and spark energy for the different piece heights.

The formation of two regions, in all different piece heights, can be seen, in which the probability curve quickly changes by small input changes. These regions, due to the intrinsic uncertainty of the input variables, should yield difficult to control regions for systems that use the ignition delay time to define the feed rate.

### 7. Validation Points

To perform the validation, four points were chosen based on the sensitivity maps and two more points. From the sensitivity maps, two of the four validation points are in the stable blue region and two points are in the yellow region.

The validation points are analyzed by plotting the observed cumulative probability of the experiment, the model's outcome, and a reverse fitting curve. The reverse fitting involves searching for the optimal gap distance and energy values for a given piece height though a numerical search. This way, it is possible to evaluate if possible deviations from the observations arise due to uncertainty in the gap distance or spark energy.

The validation points are shown in the following Table 4:

**Table 4.** Validation points and the respective uncertainty.

| Piece Height [mm] | Spark Energy [mJ] | Gap Distance [μm] |
|:---:|:---:|:---:|
| 5 | 4.8 ± 1.53 | 19 ± 1.17 |
| 10 | 5.3 ± 1.69 | 10 ± 1.39 |
| 20 | 0.89 ± 0.29 | 17 ± 1.85 |
| 100 | 10.3 ± 3.32 | 5 ± 3.78 |
| 73 | 0.89 ± 0.29 | 9 ± 2.90 |
| 73 | 17.9 ± 5.74 | 8 ± 2.90 |

The observed cumulative probability curve can be plotted alongside the model's result to visually evaluate the model.

Figures 11 and 12 display the cumulative distribution for the validation experiments (blue), the model's result (orange), and the reverse fitting with optimal values to match the observations (yellow).

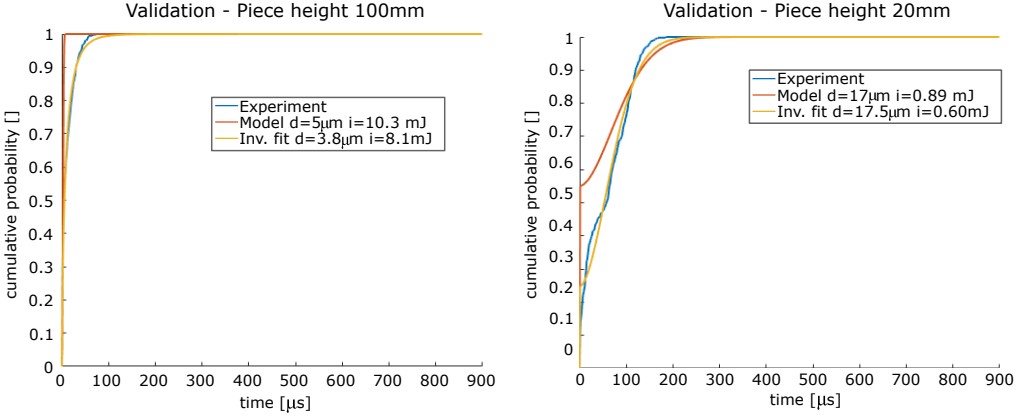

**Figure 11.** Validation of model predictions for a 100 mm and 20 mm high workpiece.

These figures show that the best fit still falls in the uncertainty of the input variables, supporting the usefulness of the model.

Even for a piece height at an unseen level, 73 mm, the model seems to capture the curve behavior and the reverse fitting results in values inside of the error of the inputs, as can be seen in Figure 13.

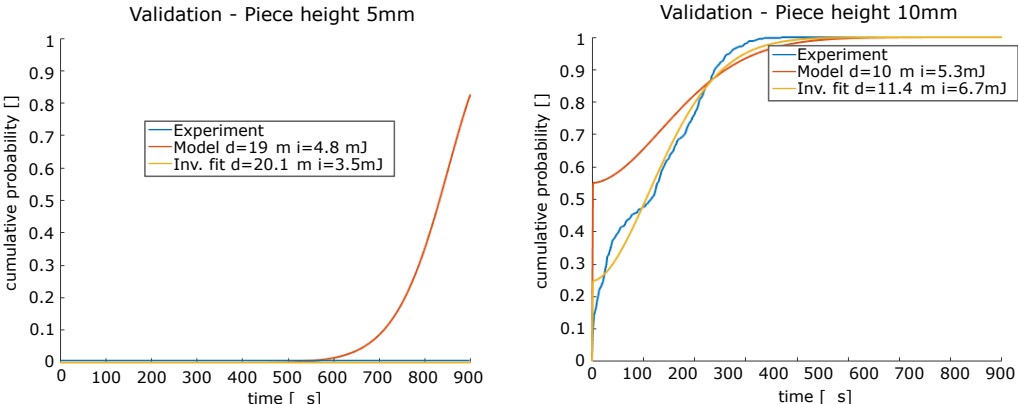

**Figure 12.** Validation of model predictions for a 5 mm and 10 mm high workpiece.

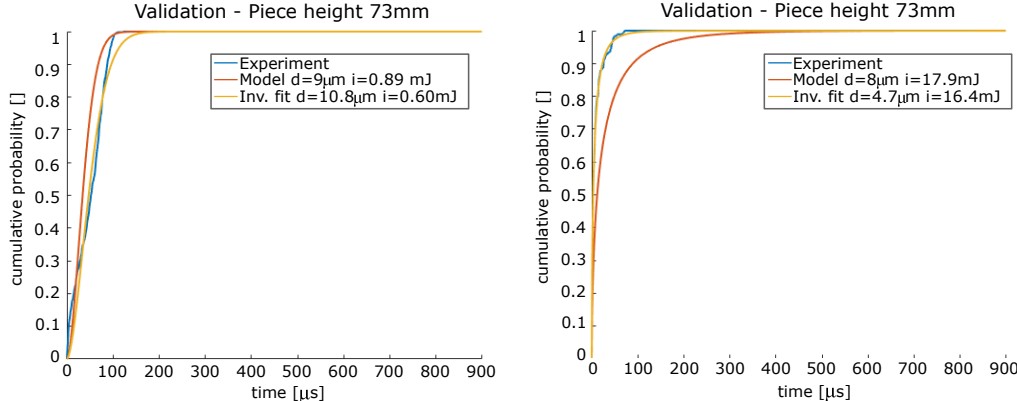

**Figure 13.** Validation of model predictions for a 73 mm high workpiece.

## 8. Conclusions

In this study, a comprehensive investigation on the ignition delay time in WEDM was conducted and a model to predict ignition delay time ($t_d$) based on various input parameters was developed. The research focused on understanding the influence of gap distance, spark energy, and piece height on the distributions of $t_d$ and developing a predictive model that captures the complex properties of these distributions.

Through extensive experimentation and data analysis, distinct patterns in the $t_d$ distributions based on the input parameters were identified. Specifically, the gap distance was found to exert a significant influence on $t_d$, with a critical gap value separating the distributions into two distinct modes. For gap distances below around 10 μm, the $t_d$ had a dominant monotonic decreasing behavior, while above this threshold, bimodal distributions prevail. This presents a practical difficulty for the control of the WEDM process and emphasizes the importance of optimizing the gap distance to enhance machining performance.

Furthermore, the spark energy exhibited erratic effects on $t_d$ distributions, with varying degrees of influence observed. The relationship between spark energy and $t_d$ was not linear or straightforward, indicating the complex nature of the EDM process and the need for careful consideration of energy settings.

Additionally, the piece height demonstrated undeniable influences on $t_d$ distributions, although without a clear overall trend. The impact of piece height on the $t_d$ varied across different parameter combinations. Further research is necessary to explore these relationships in greater detail.

To model the $t_d$ distributions, a superposition of two probability distributions was employed using two Weibull probability curves. The choice of a mixture model was

motivated by the observed bimodal nature of the $t_d$ distributions, with distinct peaks corresponding to different discharge modes.

Additionally, a sensitivity analysis was conducted, aiming to explore the regions where changes in input parameters significantly affected the $t_d$ probability curves. Through systematic variations of individual parameters while keeping others constant, the most influential factors driving the model's response were identified. The resulting sensitivity maps provided critical insights into the behavior of the system and revealed regions of difficult feed rate control. Small changes in input parameters led to significant variations in $t_d$. These findings highlight the challenges associated with using $t_d$ for defining feed rates in practical applications and emphasize the importance of careful parameter selection and control in EDM processes.

In conclusion, this study has contributed to a deeper understanding of the ignition delay time in WEDM and has presented a modeling approach for predicting $t_d$ based on gap distance, spark energy, and piece height. The developed model can serve as a valuable tool for optimizing WEDM processes and determining machining parameters that should avoid unstable cutting conditions, if the $t_d$ is used to guide the feed rate.

Future research endeavors could focus on explaining the effect of the machining area on the $t_d$ or refining the model by incorporating additional input parameters, investigating the influence of other process variables on the $t_d$. Another interesting area would be the use of such models for simulating the cutting process.

**Author Contributions:** Conceptualization, P.M.B.E.; Formal analysis, P.M.B.E.; Investigation, P.M.B.E. and M.H.; Methodology, P.M.B.E. and M.H.; Project administration, M.K. and K.W.; Resources, M.K. and K.W.; Supervision, M.K. and K.W.; Validation, P.M.B.E. and M.H.; Writing—original draft, P.M.B.E. and M.H.; Writing—review and editing, M.K. and K.W. All authors have read and agreed to the published version of the manuscript.

**Funding:** My research has been institutionally funded by ETH Zürich.

**Data Availability Statement:** The data presented in this study are available on request from the corresponding author. The data are not publicly available due to privacy.

**Conflicts of Interest:** The authors declare no conflict of interest.

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
