# Peer review of "Ignition Delay Time Modeling in Wire-EDM"

_jmmp, doi:10.3390/jmmp7050177_

Round 1
Reviewer 1 Report
How is the discharge gap measured?
Reviewer 2 Report
The article discusses the ignition delay time modeling in WIRE-EDM. The article is well written and well structured. The presentation of the material is logical. The quality of the drawings is high. A few comments on the article.
1. The ‘Introduction’ section is not comprehensive and well-organized enough. Research background and recent development of similar studies should be reviewed, and research methods, innovation and significance of your paper should be stated. Please enrich the ‘Introduction’ section.
2. Using a list of lumped references is not very helpful for a reader (for example, ... [1-4] on page 1 or [5,6,8,10,11] on page 1). Assessment/justification should be provided for each reference, even it may be short.
3. I believe that the "Notation and Abbreviations" section of the article is necessary for the convenience of readers.
4. It is necessary to add a comparison of the results obtained with the data of other authors.
5. It is necessary to indicate directions for further research on this topic in the "Conclusion" section.
In my opinion this article is interesting and worthy. However, the article has a number of shortcomings. To accept this paper for publication in the J. Manuf. Mater. Process., some improvements and revisions are required as specified.
Reviewer 3 Report
The study aims to model the ignition delay time (td) in electrical discharge machining (EDM). While the topic and approach are interesting, several concerns are raised about the novelty, generalizability, and applicability of the research:
-
The paper fails to highlight significant novelty or advancements compared to existing studies in the field. EDM mechanisms and discharge delay have been thoroughly studied for at least two decades, and the current work does not seem to offer any substantial new insights or improvements.
-
The motivation for the study is not well-defined, and the projected impact on wire EDM process optimization is not convincing. Many studies and models, including AI, empirical, and analytical approaches, already exist in this area, rendering the potential impact of this study questionable.
-
The presence of Table 1, summarizing tolerance deviations specified by the machine manufacturer, raises concerns about the study's specificity to a particular machine. It is unclear whether the model can be applied to different machines, wire materials, wire diameters, workpiece materials, and dielectrics. Without this generalizability, the practical application of the findings becomes limited.
-
Table 2 indicates specific ranges for the workpiece height, potentially limiting the model's applicability to other workpiece sizes and shapes. Additionally, the validation points chosen for the model are also very specific, further raising questions about the model's reliability and accuracy under different conditions. What if the workpiece is 200 mm high (there are wire EDM machines capable of machining even thicker workpieces)? Also, will be model holds well for continuously variable workpiece heights, such as triangular workpieces? Such omissions suggest that the model's robustness may be limited to certain predetermined scenarios, making it less useful for real-world applications.
Overall the language is fine.
Reviewer 4 Report
This study examines and models the ignition delay time (td) in electrical discharge machining (EDM). It investigates how gap distance, discharge energy, and piece height impact the stochastic distributions of td, offering valuable insights into their intricate properties. Although it is a good work, I have the following concern with regard to the paper which need to be addressed:
1. The results and discussion sections present impressive results; however, a notable drawback is the need for more discussion and comparison with similar published literature. To enhance the study, it is crucial to include an in-depth discussion that highlights the novel discoveries and offers meaningful comparisons with existing research.
2. For the abstract section, it would be beneficial to include specific quantitative results or trends observed in the study. In addition, it is crucial to highlight any significant findings in the modelling approach used for predicting the td. Providing these insights will offer readers a better understanding of the study's findings and their significance.
3. More strong keywords are required to attract the readers and for easy search of the article.
4. If the work presents a study on WEDM, then why is it EDM writted in the abstract?
5. I found number of articles on the effect of discharge energy and gap distance on WEDM. For instance: 10.1080/10426914.2020.1854462, 10.1080/10426914.2023.2176875, 10.1007/s00170-022-10608-2, 10.1080/2374068X.2022.2079590, etc. What is new in this work?
The references cited are obsolete (1990 onwards), I suggest the authors to add recently published articles (2010 onwards) in the context of the work to gain visibility.
6. What type of power supply was used in the experimental setup?
7. What kind of pulse were generated in the pulse generator?
8. What was the surface roughness of the substrate before machining? How were the surface roughness evaluated? Parameters considered for the measurement conditions?
9. How was the repeatability ensured? How many set of experiments conducted in the study? How many evaluations were taken in the study for each parameter measurement?
10. How was the data acquisition done?
11. Figure 4 need to be enhanced.
12. Conclusion: It needs to be shorter and more concise. The authors can provide how the findings of this study have advanced the previous existing results by comparison. This would help to indicate the novelty of the work more clearly. There must be an inclusion of future scope of study (if any) in this part.
Round 2
Reviewer 2 Report
The authors took into account all comments. I believe that the article can be accepted for publication.
Reviewer 4 Report
The authors have addressed the said comments and the paper looks fine to me now.